# Enhancing Failure Mode and Effects Analysis Using Auto Machine Learning: A Case Study of the Agricultural Machinery Industry

**Sami Sader** [1,*] , **István Husti** [2] **and Miklós Daróczi** [2]

1   Doctoral School of Mechanical Engineering, Szent Istvan University, 2100 Godollo, Hungary
2   Institute of Engineering Management, Szent Istvan University, 2100 Godollo, Hungary;
    husti.istvan@gek.szie.hu (I.H.); daroczi.miklos@gek.szie.hu (M.D.)
*   Correspondence: sami.s.a.sader@phd.uni-szie.hu; Tel.: +36-702235922

**Abstract:** In this paper, multiclass classification is used to develop a novel approach to enhance failure mode and effects analysis and the generation of risk priority number. This is done by developing four machine learning models using auto machine learning. Failure mode and effects analysis is a technique that is used in industry to identify possible failures that may occur and the effects of these failures on the system. Meanwhile, risk priority number is a numeric value that is calculated by multiplying three associated parameters namely severity, occurrence and detectability. The value of risk priority number determines the next actions to be made. A dataset that includes a one-year registry of 1532 failures with their description, severity, occurrence, and detectability is used to develop four models to predict the values of severity, occurrence, and detectability. Meanwhile, the resulted models are evaluated using 10% of the dataset. Evaluation results show that the proposed models have high accuracy whereas the average value of precision, recall, and F1 score are in the range of 86.6–93.2%, 67.9–87.9%, 0.892–0.765% respectively. The proposed work helps in carrying out failure mode and effects analysis in a more efficient way as compared to the conventional techniques.

**Keywords:** Industry 4.0; auto machine learning; failure mode effects analysis; risk priority number

## 1. Introduction

Failure modes and effects analysis (FMEA) is a proactive analytical technique for identifying, tracking and mitigating product and process potential failures in a systematic way by determining its potential occurrence, root causes, consequences, and impact [1]. FMEA provides a quantitative score to evaluate failures where every failure is transformed into a numerical value that is called risk priority number (RPN). RPN is the result of multiplying three parameters namely severity, occurrence, and detectability. Severity is the risk or damage that may affect the machine, product, next operator or the end-user. On the other hand, occurrence is the likelihood of this failure that may occur again. Finally, detectability is the degree to which this failure could be detected [2–4]. Higher RPN value represents a higher priority of risk [5]. Appropriate corrective actions are usually determined based on RPN threshold value. If this threshold is reached, a risk mitigation procedure is applied accordingly [6]. Moreover, RPN value is used as a tool for optimal resource allocation by giving focus on risks that have the highest RPN or the most critical issues [3,7].

FMEA was firstly developed by NASA in 1963 to enhance the performance of the devices that are used in the aerospace industry [8]. Later, FMEA was adopted and promoted by Ford Motors in 1977 [3]. Currently, FMEA is being used in the automotive industry to ensure the quality and reliability of production systems [9]. Daimler Chrysler, Ford, and General Motors have developed an international

standard called SAE J1739_200006 as general guidance for implementing FMEA techniques to avoid failures and enhance system reliability and safety [10]. FMEA documents are classified into two types namely design FMEA, and process FMEA [11]. Design FMEA is constructed during product design to define product weaknesses, critical components and their respective potential failure modes, root causes, and effects [1]. Meanwhile, process FMEA focuses on potential failures that may occur during the manufacturing process and incurred risks at each process step [3].

FMEA is a robust tool for quality improvement in both manufacturing and services industries. It can be used at the design stage of the product and during its implementation [9]. The aim of this is to avoid the end-user from experiencing unfavorable defects that may affect the reputation of the company negatively [3]. FMEA is also used as a process improvement technique to ensure consistency, reliability and avoid deviations. Moreover, it is also used to define and mitigate risks [12]. On the other hand, FMEA is used to improve maintenance management by analyzing the maintenance requirements of the product and developing the maintenance plans that would be used to ensure that the system is doing what it is meant to do when it was created. Finally, FMEA is used to improve safety by conducting hazards analysis of components that have critical hazards on lives, property, or other losses that are identified and mitigated [7].

However, FMEA is criticized for many conceptual aspects. The most popular disadvantage of this method is the narrative and qualitative nature of its structure. For every product or process, FEMA documents are developed by engineers and experts using linguistic terms that are based on the personal evaluation. The RPN parameters' values are determined by engineers and experts which may include uncertainty and vagueness [12]. Moreover, the parameters that are used in FMEA are represented by (1–10) crisp scale which is an unreliable representation of real-application cases [5,13]. Additionally, Chang, et al. in [3] have criticized the RPN estimation by the inhomogeneous morphologic correlations between the three parameters. This criticism is based on the fact that each of these parameters is obtained and linearly multiplied by the other with an identical scale. This process is done despite the actual impact of every independent parameter and the different qualitative interpretation of the scale. For example, high severity value should result an extremely high RPN value due to the critical hazard on the operator or the machine. In other words, once there is a risk on human, the other parameters shouldn't downgrade the overall value of RPN even if they are low.

Thus, in order to overcome this ambiguity, researchers proposed several approaches to improve the application of FMEA and the development of RPN. Several fuzzy techniques were examined to develop a new risk assessment approach to overcome the weaknesses of FMEA. Haktanır and Kahraman in [13] have summarized several fuzzy techniques and grey theory and proposed interval-valued neutrosophic (IVN) sets-based FMEA to eliminate the inaccuracy of human decisions and evaluations. Ayber and Erginel in [12] have proposed single-valued neutrosophic (SVN) Fuzzy FMEA as a new risk analysis tool to overcome the ambiguity of the linguistic terms. Al-Khafaji, et.al in [14] have proposed a fuzzy multicriteria decision-making model aligned with FMEA principles to obtain an efficient criterion for maintenance management. Liu et al. in [5] have used cloud model theory and hierarchical TOPSIS method to enhance FMEA effectiveness, overcoming bias probability of human judgment, and to facilitate the transformation of qualitative terms to quantitative values. Yang et al. in [2] have utilized a data mining-based method for isolating faults based on FMEA parameters in order to enhance predictive maintenance by using historical big-data to create data-driven models, by which future failure can be predicted efficiently and accordingly avoid failures at a very critical operational item. Keskin and Özkan in [6] have applied a fuzzy adaptive resonance theory (ART) method for FMEA modeling in order to improve the classical methodology of calculating the RPN, which in total minimized cost and efforts needed to respond to corrective actions alerts.

In the aforementioned research, the interpretation of FMEA documents was well addressed and resolved. However, the weakness of FMEA and RPN is not limited to the ambiguity of the FMEA textual description nor its quantitative representation, but it also extends to the importance of being proactive and responsive to failures. The flow of information once a failure is detected until the time it

is ranked and resolved is important as well to guarantee minimum impact and limited implications. Another shortcoming of the conventional FMEA technique comes from the fact that its documents are prepared during the product or process design stages, which makes these documents obsolete after production starts ahead. Therefore, these documents need to be dynamically validated and updated on a continuous basis. Hence, utilizing new technologies is very vital to overcome these weaknesses and keep these documents updated and responsive [2].

In the era of Industry 4.0, connectivity offered instant communication and collaboration among the value chain. Artificial intelligence (AI), the internet of things (IoT), big-data, and cyber-physical systems (CPS) made a great leap in automation and optimization at all levels of manufacturing. Here, automation is not limited to machines and processes, but also to management information systems such as enterprise resources planning (ERP), customer relationship management (CRM) and quality management systems (QMS) [15]. Additionally, the real-time flow of data among the value chain, which is instantly analyzed and transformed to user-friendly information, thanks here to the advanced supercomputing and analyzing power [16], resulted new paradigms of manufacturing systems which are being called nowadays by smart factory, smart machine, smart product and augmented operator [17]. These pillars changed the production systems from being reactive to be proactive and levered the human intervention from doing the work to supervise it while it is being done. Sensors, 3D cameras, radio frequency identifier (RFID), and Wi-Fi made monitoring processes more precise and accurate. Unseen defects or deviation of products or processes can be detected as soon as it is occurring. Defect elimination and processes re-adjustment are made autonomously at the micro and macro levels [15,18,19]. All these technologies, alongside the increased complexity of products and their manufacturing systems, generated a large volume of data, at a high velocity, veracity, and variety. The analysis of such big data requires advanced resources and techniques to classify data and detect patterns that cannot be detected using traditional analytical tools.

Automated machine learning (AutoML) are tools that automate the process of a machine learning workflow, offering the same capabilities of regular machine learning, without explicit knowledge of programming [20]. AutoML aims at reducing the human intervention in data preprocessing, feature selection and algorithm selection so as to make machine learning automated [21]. Google AutoML is a cloud machine learning platform that automates supervised machine learning in a very efficient way. It handles the tasks of data preprocessing, feature extraction, feature engineering, feature selection, algorithm selection, and hyperparameter optimization [22]. Google AutoML automatically develops models based on neural architecture search (NAS). It follows the try and error strategy by developing the model based on a random set of hyperparameters, then evaluates the performance of the model which is resulted by using this set of hyperparameters and finally concludes the most accurate model [22,23].

AutoML is increasingly used in scientific research areas. Faes et al. in [24] have evaluated the performance of AutoML hosted by google cloud platform against other machine learning methods and algorithms. It is claimed that AutoML has higher accuracy in medical image classification and can be used by people who are less experienced in coding and algorithms. Similarly, Hayashi et.al. in [25] utilized Google AutoML to identify pest aphid species and improving crop protection effectiveness. The authors concluded that such a tool provided an accuracy of 0.96 which allowed them to consider the AutoML as a useful and effective tool. Additionally, Li et al. in [26] have used AutoML to automate customer service activities by analyzing different customers' information and respond to their inquiries based on historical frequent inquiries. According to the authors, the solution provided improved responsiveness and minimized the cost of customer service management. Moreover, Galitsky et.al in [27] proposed a novel approach to automate customer complaints processing and classification by training a machine learning algorithm for analyzing dialogues recorded between customers and company-agents.

Based on that, the aim of this paper is to examine a novel optimization approach applied to FMEA and RPN by classifying failures according to updated FMEA documents and generating the

RPN automatically without human intervention. A successful FMEA is gained through optimized consistency, responsiveness, and accumulated experience. The suggested approach aims at solving the above-mentioned FMEA weaknesses by two steps: first, reviewing and re-evaluating a dataset containing reported failures manually by experts to ensure accuracy. Secondly, conducting supervised machine learning techniques on the updated data and develop machine learning models that can be deployed to evaluate and classify newly reported failures automatically with minimum processing time and enhanced consistency.

## 2. Study Background

In this research, CLAAS Hungária Kft (CLH) is adapted as a case study. CLH was established in 1997 in Hungary as a subsidiary company of CLAAS Group. CLAAS group is an international German family-owned business company based in Germany and owns many manufacturing plants worldwide. CLAAS is a world-leading manufacturer of agricultural equipment and machinery such as tractors and combine harvesters. Since establishment, CLH expanded from 350 workers and 8 hectares plant to more than 700 workers working on a 14-hectare plant and became a center of excellence for combine harvester tables and trolley carts production. CLH manufactures supplementary devices such as combine harvester tables, cutting heads, and trolley carts, as shown in Figure 1. These devices are shipped from Hungary either to the mother company that is located in Germany or directly to the end-customers for final assembly with the machine which can be a combine harvester or a tractor. The cost of a single failure is tremendously high, not only due to the machine cost itself but also due to the entailed logistics and the re-work cost.

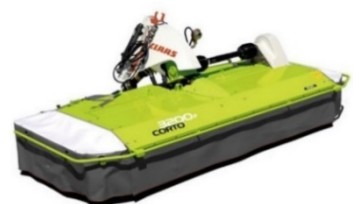 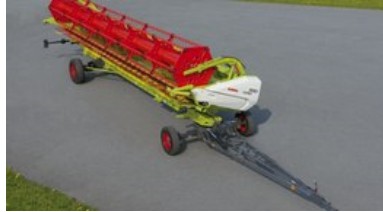 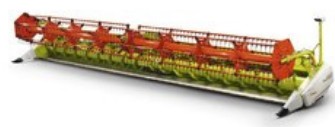

**Figure 1.** Sample of devices manufactured at the subsidiary company subject of this study.

CLH's staff has developed "Quality Checklists" for every product, process or manufacturing phase. These quality checklists are developed based on the FMEA documents and are being used at the quality gates in the shop floor in order to ensure that common failure causes are avoided. Moreover, this process aims to make sure that critical device components are installed and configured at the optimal conformance to design. However, as mentioned earlier, FMEA documents are prepared during the product design phase and can be changed once the serial production is initiated. Meanwhile, further failure modes can be detected at the final assembly phase. Therefore, these quality checklists are demanded to be dynamic, updatable and responsive to real quality issues reported during or after production.

This research activity is focusing on a single device that consists of the combine harvester feeder house as shown in Figure 2. Feeder house is a device that is attached to the combine harvester to facilitate the control of the cutting head and the flow of crops from the cutting head to the combine harvester. The device consists of several complex systems such as mechanical, hydraulic, electrical, and electronic systems. This device is wholly manufactured in the subsidiary company in Hungary and dispatched to be assembled to the combine harvester at the mother company in Germany.

Failures or defects which are observed during assembly or reported by end-users are gathered on a daily basis through the global ERP system of the company. After that, this information is extracted and manually and reviewed by an experienced quality management team. This evaluation process aims at analyzing root cause and consequently taking the needed correction actions in order to maintain profitability and high-quality production. The company uses an internally customized

FMEA technique to evaluate reported claims by obtaining RPN for every claim according to FMEA documents. The method which is used here aims at generating an RPN value for every claim on a scale from 1 to 300 points, where 300 is the highest priority number.

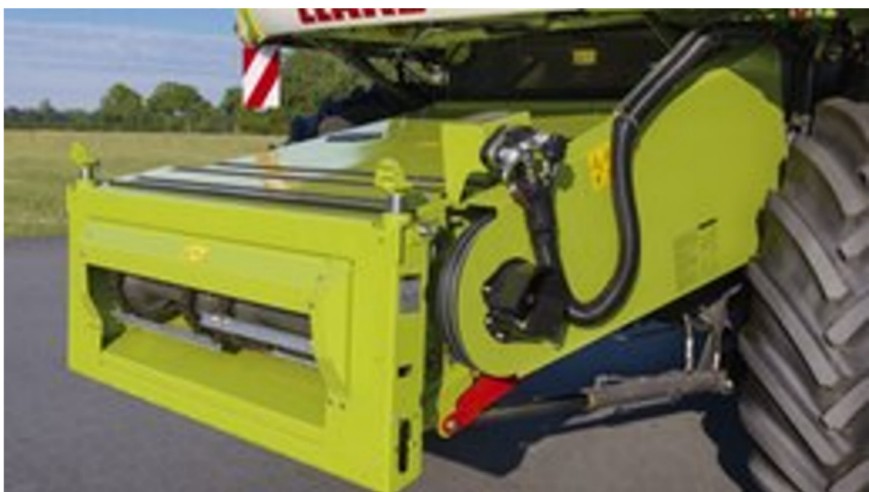

**Figure 2.** Feeding house attached to the combine harvester body and ready to be attached with a cutting head.

RPN in this CLH is obtained based on three major factors: (severity, occurrence, and impact). Severity, or gravity as named by the company's internal manuals, represents the risk consequences of the claim from customer and company perspectives. It also includes the cost of resolving this issue and the safety impact on the operator. The weight of this factor ranges between 1 to 10 points, where 1 is the lowest severity and 10 is the highest. In the meanwhile, occurrence represents the number of incidents a specific claim has been witnessed in a specific period. The weighting scale of this factor is also 1–10, where 10 is the highest. Impact is weighted by a scale of 3 points from 1 to 3. Impact represents the repair efforts, time, repetition of the same work, and the overall impact of the claim on the reputation and image of the company. The meaning of every scale value from 1 to 10 is elaborated in detail in [3,9]. The evaluation process is summarized in Figure 3 below.

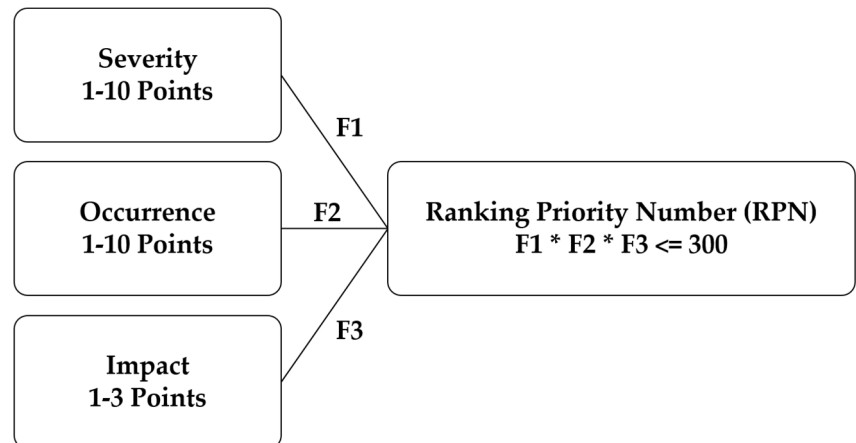

**Figure 3.** Factors affecting claim ranking and the weight of every factor.

Equation (1) shows the multiplication of the three factors values that results an RPN value between 1 and 300 points. An RPN value above 160 points is classified at a very high priority, while, a value

between 100 and 160 points is classified as a high priority. Medium priority is noted if the RPN value is in the range of 35–100, while low priority is noted if the RPN value is less than 35:

$$RPN = Severity \times Occurrence \times Impact \tag{1}$$

According to the RPN value of every claim, the quality team decides the next handling steps. Further steps could be tracing root cause(s) and ensuring the elimination of such cause(s) and/or updating the quality checklists to ensure further failures will not repeat in the future. Time and experience play a crucial role in this regime. It is important to improve the process of evaluating claims and lever the current experience.

The evaluation and ranking process requires highly experienced people who are fully aware of the FMEA documents and its applications. The volume, velocity, and veracity of claims reported, and their processing time is very critical from a quality management perspective. It is essential in such a high-value industry to resolve issues as soon as they are reported. Early and fast processing of quality issues is translated to a lower quality cost and will positively enhance the general business performance. Moreover, standardization of the evaluation process and consistency of the process is vital to guarantee consistent RPN results every time.

The accumulated experience, time of processing, consistency of the evaluation process can be attained through the proposed solution in this paper; utilizing automated machine learning to classify and analyze claims data. Machine learning capabilities provide the capacity to analyze several input features (columns) at one dimension, aligned with a large volume of data (rows) at the other dimension. This helps in discovering and analyzing unseen factors, considering that the best quality practices focus on the claim root cause analysis. Additionally, utilizing technology whenever possible is very promising in the industry, because of its availability at any time (24/7) under any conditions and its ability to go deeper in analysis beyond human capacity. Delegating such tasks to machines will let human intelligence focus on higher strategic issues and to reach a higher level of efficiency and effectiveness.

## 3. The Proposed FMEA Analysis Method

In this section, it is suggested to utilize supervised machine learning technology to replace human intervention in processing, evaluating, and categorizing claims. The current flow of claims from involved parties is illustrated in Figure 4. Claims from internal company quality product audit (product audit claims) and issues that were detected during assembly (cross-company claims) are pipelined in the company's ERP system and human intervention is important at one point to evaluate claims manually. Based on the evaluation results, quality management decides how to deal with every single claim to find the root cause of the problem. This is done either by following the eight disciplines of problem-solving (8D) methodology for critical or high-ranking issues or by just updating the shop floor quality checklists in order to ensure the quality of next produced devices. Otherwise, this reported issue is just as it is an accidental incident and occupies a very low RPN value.

Accordingly, a dataset that contains one-year data of claims is extracted from the ERP system of the company. This data is concerning the selected device only (the feeder house shown in Figure 2). Firstly, to ensure the accuracy of the developed models, the data was re-evaluated and validated manually by experienced quality engineers to obtain the three RPN elements (severity, occurrence, and detectability) and to define the root cause and the source manufacturing process (such as cutting, bending, welding, painting, assembly, etc.) of every claim. The evaluation process depends on the experience of the quality team and based on the internal FMEA procedure for every failure mode. The resulted updated dataset is used for training and develop an ML model that is deployed to predict an RPN value for future failures claims and classify its root cause instantly without further human intervention.

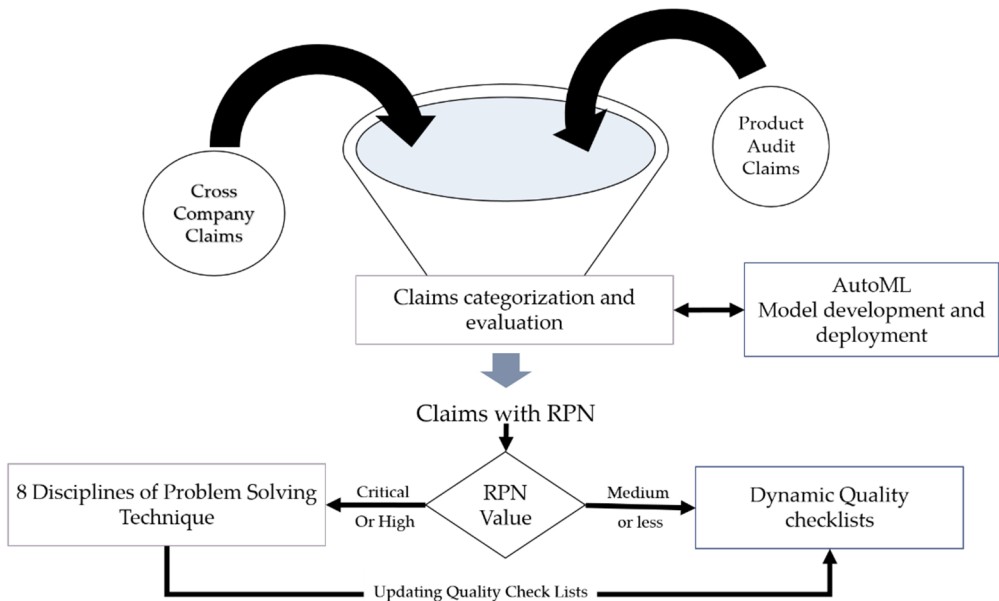

**Figure 4.** The flow of internal quality audit and cross-company claims to quality management.

In this research, Google AutoML is selected for three reasons: first, its effectiveness, as Google AutoML is free to try and utilizes the latest ML technology developed by Google brain team. Second, its ease of use, which is very important to ensure the sustainability of the project results after the research cooperation ends, keeping in mind that people who are working at the partner company are less experienced with coding and modeling. Therefore, such a friendly system will ensure that the partner company can deal with the work after the end of the project with the least knowledge of coding and data processing. Third, its ability to integrate. It is agreed with the partner company to integrate the developed models in the company's ERP system. Google AutoML offers the ability to deploy and integrate the resulted models by application programming interface (API).

Supervised machine learning is used to improve the failure claims processing process. Claims are reported by engineers at the assembly location in Germany or from service centers to the quality management office through the company's ERP system. Claims are analyzed, categorized and ranked by the quality management team based on FMEA documents. Accordingly, failures are ranked and prioritized based on their importance and critical impact.

The proposed solution aims at developing an automatic claim ranking system to replace human intervention based on developing four machine learning models that can read, analyze, evaluate, and assign relevant ranking values for every processed claim. In order to do so, a dataset of already evaluated claims is used to train the model. Afterward, the model will be deployed to evaluate new claims based on the experience gained by the training data. Figure 5 below elaborates on the process of models' development, its inputs, and outputs. The first step in models' training is to preprocess the input data, feature selection, and data types. The auto machine learning tool results four models that will be able to predict four independent values by which three of them will be multiplied to calculate an RPN value. The fourth decides the source manufacturing process of the same claim.

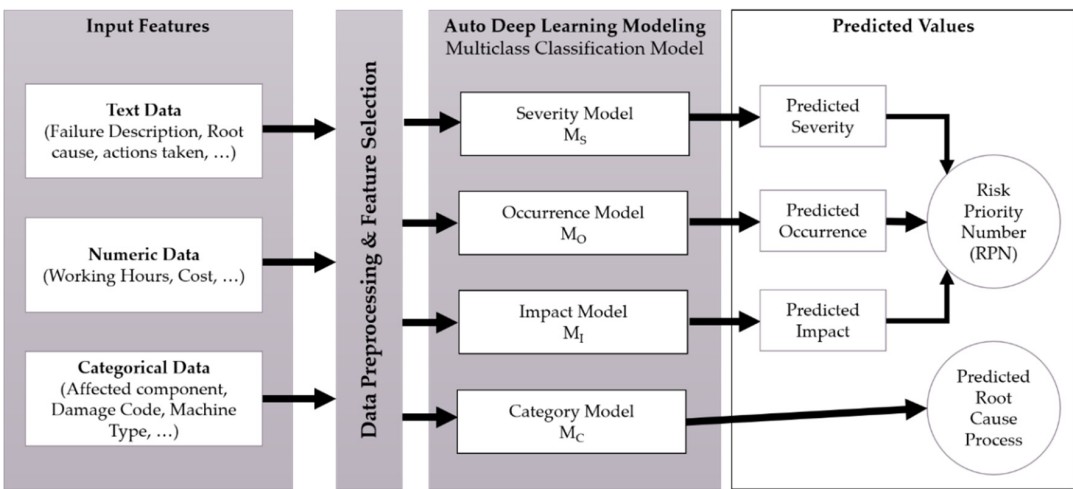

**Figure 5.** Development of machine learning model.

### 3.1. Data Pre-Processing

A dataset that contains 1532 rows of failure incidents has been received from the industrial partner of this research project. The data was extracted from the ERP system of the company, it was recorded over one year, and related only to the selected product in Figure 2. Every row in the dataset contains the details of a single incident and described by 23 different input features (columns) that help the quality engineers to recognize the failure mode and therefore, refer to the FMEA documents to assign the proper RPN value that fits this failure mode. For example, a failure is claimed from the assembly line in Germany where the engineers reported an incident of "an insufficiently tightened screws at one component in the device", along with this reported failure, further information are provided such as the serial number of the device, the code number of the component as in the design, further description written in textual format by the labor who solved the issue including his opinion on the issue and its criticality, the damage code as picked from the list of options in the input screen, the expected root cause of the problem is explained in text, the time consumed to fix the problem, and cost involved for rework, and the final conclusion. Table 1 summarizes the input features types and roles in the models. Whereas this dataset is used to develop the machine learning models. The first step is to prepare the data for the AutoML platform. This includes ensuring that all features of the dataset are organized, and data types are well defined. Additionally, the claims are validated manually by the quality management team at the partner company using a specially programmed interface that facilitates the manual validation process. This manual validation of data was made in order to ensure the quality of the input data and therefore ensure the quality of the output models.

**Table 1.** Dataset input features for the machine learning model.

| Data Type | Number of Inputs | Labels | Brief Summary |
|---|---|---|---|
| Textual Text | 3 | Claim description, Root cause, and remediation action made | • This data is written in natural language by the labors or engineers at the German company, explains the failure, its root cause, and the remediation action made<br>• It will help to recognize the failure mode, its root cause, and its technical solution |
| Categorial Data | 10 | Machine code and name, damage name and code, initial criticality assessment, component type, and reporter information | • Contains data about the device affected, the damage category, and its criticality<br>• It will help to identify reoccurrence of similar failure, evaluate its importance, and define the location at which it was detected |
| Numeric Data | 10 | Different costs data, number of affected devices | • This data will help to evaluate the consequences of this failure in terms of labor cost, transportation, material cost, and any extra costs |

Furthermore, 46 rows are excluded from the training process because of missing critical details such as claim textual description and the root cause input. Moreover, scales (8–10) in severity and (7–10) in occurrence had an insufficient number of claims (lower than 50 rows) for every element, these records are excluded too, as shown in Table 2a. The reason behind that, AutoML platform cannot start training with less than 50 readings per class. Therefore, the dataset is copied three times, and classes with less than 50 readings are eliminated. Finally, 1484, 1425, and 1486 claims are used for models training of severity, occurrence, and impact respectively. The data plot is shown in Figure 6 where the distribution of the data is illustrated.

**Table 2.** Summary of dataset included in the modeling.

| a. FMEA Elements | | | |
|---|---|---|---|
| **Scale** | **Number of Records** | | |
| | **Severity** | **Occurrence** | **Impact** |
| 1 | 182 | 267 | 866 |
| 2 | 454 | 199 | 511 |
| 3 | 167 | 424 | 109 |
| 4 | 291 | 204 | |
| 5 | 218 | 128 | |
| 6 | 81 | 203 | |
| 7 | 91 | 28 | |
| 8 | 0 | 10 | |
| 9 | 2 | 0 | |
| 10 | 0 | 23 | |
| Total | 1486 | 1486 | 1486 |
| Dataset rows | 1484 | 1425 | 1486 |
| Training rows | 1343 | 1269 | 1355 |
| Evaluation rows | 141 | 156 | 131 |

| b. Category of Claim | |
|---|---|
| **Process Category** | **Number of Records** |
| Category A | 89 |
| Category B | 104 |
| Category C | 464 |
| Category D | 123 |
| Category E | 206 |
| Category F | 86 |
| Category G | 303 |
| Category H | 68 |
| Others | 43 |
| Total | 1486 |
| Dataset rows | 1443 |
| Training rows | 1309 |
| Evaluation rows | 134 |

In addition to RPN evaluation, the research work includes classification of claims according to the respective manufacturing process which is described to be the root cause process of the defect. The names of processes are masked in Table 2b where the process could be any of the known machining processes such as cutting, bending, welding, assembly, etc. Processes with less than 50 records are excluded as well.

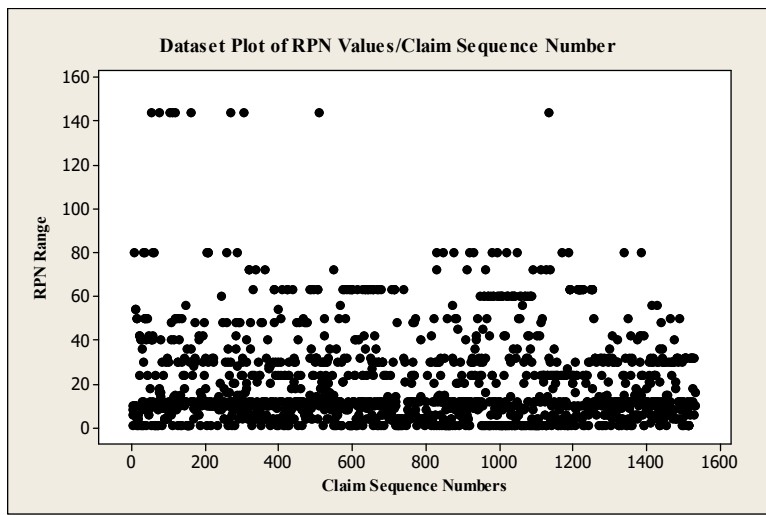

**Figure 6.** Dataset plot of all claims based on RPN value.

## 3.2. Data Modeling

The AutoML platform analyzes the dataset and constructs the models automatically. Such analysis includes text processing is made by the Google AutoML in addition to the other categorical and numeric data.

Multiclass classification technique is applied to develop four Machine Learning models, while the suitable algorithm is automatically developed by Google AutoML. Google AutoML is developed to help researchers in handling large data and building high accuracy models with the least coding experience and resource consumption. The datasets are uploaded, the input features are defined, and targeted elements are selected. The prepared datasets are analyzed to obtain three independent models that can evaluate every claim to predict three independent values for severity, occurrence, and impact, from which an RPN can be calculated by applying Equation (1). Additionally, a fourth model (for category) is obtained to identify the manufacturing process which caused this failure to occur. The manufacturing process could be cutting, bending, welding, painting, assembly, packaging, and transportation. The aim of the fourth model could be extended in the future to include more specific processes such as welding machine 1, assembly line 2 and so on. Figure 7 illustrates the four models obtained after training. As the AutoML platform is a cloud system, then the consumption processing can be measured by node hour. The training process consumed 0.944, 1.105, 0.86, and 1.111 node hours for severity, occurrence, impact, and category respectively. Every node hour includes the use of 92 n1-standard-4 equivalent machines in parallel, where a single n1-standard-4 machine operates four virtual CPUs and 16 GB of RAM memory.

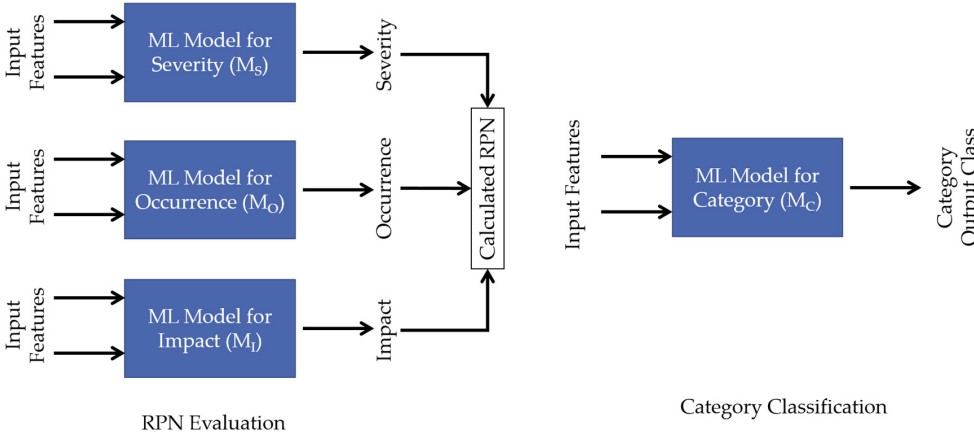

**Figure 7.** RPN evaluation and Category Classification models.

### 3.3. Models Evaluation

The models' evaluation is done by three evaluation metrics namely precision, recall, and F1 score. Further evaluation metrics are adopted here such as area under curve (AUC) and the confusion matrices.

Precision is the percentage of true positive predictions to all positive predictions (true positive and false positive). Recall is the percentage of true positive predictions among all actual values (true positive and false negative). While F1 score is a balanced evaluation between precision and recall, and it is used especially when the data in the datasets are not equally distributed over classes.

The area under the precision-recall curve (AUC-PR) and the area under the receiver operating characteristic curve (AUC-ROC) are used to visualize the performance of the models. AUC-PR shows the trade-off between precision and recall for the model. AUC-ROC shows the trade-off between true positive rate and false-positive rate.

The confusion matrices are used here to elaborate on the prediction accuracy and accepted tolerance of every class. For example, higher prediction resulted by any of the RPN elements models ($M_S$, $M_O$, and $M_I$) can be accepted because assigning higher ranking value to an incident increases its priority. The limitation here is the degree of tolerance accepted by the company. To elaborate more, when an incident is evaluated for a severity class of three, it is accepted if the model predicts a value that is higher than the actual value by one step. However, it could be inefficient if the model predicts two or higher steps than the incident deserves.

Finally, the accuracy of the models is affected by the type of every element, the number of rows that are used for training, the accuracy of details provided per row, and the scale of every element (or number of classes per element). It is important here to recall the objective of this work which is to provide a proof-of-concept that machine learning is an effective technique to enhance FMEA and the development of RPN value.

## 4. Results and Discussion

In this research, four machine learning models are trained and evaluated successfully in this research. Table 3 summarizes the training evaluation results and accuracy metrics for four models of severity ($M_S$), occurrence ($M_O$), impact ($M_I$), and category ($M_C$). The evaluation sample was automatically split and tested by the AutoML platform.

The performance metrics in Table 3 shows relatively high-quality models, with different levels of precision for each model. The area under the precision-recall curve (AUC-PR) and the area under the receiver operating characteristic curve (AUC-ROC) are close to 1, which indicates high-quality classification models. Moreover, the models' precision rates are 93.2%, 87.6%, 89.9%, and 86.6% for $M_S$, $M_O$, $M_I$, and $M_C$ respectively, which indicates that the models predicted correctly the classes of the validation sample for every model.

**Table 3.** Models Evaluation.

| Dataset Targeted Value | Validation Sample | Score Threshold | Precision | TPR (Recall) | F1 Score | AUC (PR) | AUC (ROC) |
|---|---|---|---|---|---|---|---|
| Severity ($M_S$) | 141 test rows | 0.5 | 93.2% (96/103) | 68.1% (96/141) | 0.787 | 0.895 | 0.970 |
| Occurrence ($M_O$) | 156 test rows | 0.5 | 87.6% (106/121) | 67.9% (106/156) | 0.765 | 0.871 | 0.955 |
| Impact ($M_I$) | 131 test rows | 0.5 | 89.9% (116/129) | 88.5% (116/131) | 0.892 | 0.954 | 0.973 |
| Category ($M_C$) | 134 test rows | 0.5 | 86.6% (103/119) | 76.9% (103/134) | 0.814 | 0.877 | 0.972 |

The highest F1 score is recorded for $M_I$, where the full dataset is used for training, and the classification was only among three classes (1, 2 or 3) while the training dataset for $M_I$ contains 866, 511, 109 readings for every class from 1 to 3, respectively.

The confusion matrices shown in Tables 4–7 below show that the concentration of the true predictions is at the diagonal cells of all models. However, both models $M_S$ and $M_O$ show higher confusion for predicted labels against true labels, in contrast to $M_I$ and $M_C$ models where higher concentration is shown at the diagonal cells. This is highly connected with the data volume and will be improved when a larger volume of data is used for model upgrading.

However, predicting a higher value than the true one (negative true predictions in the confusion matrices) for the three models ($M_S$, $M_O$, and $M_I$) could be accepted, as higher prediction value for severity will increase the RPN value and therefore, the priority to resolve the failure is increased. However, this tolerance is not acceptable for $M_C$ as it deals with a totally different interpretation, it describes the manufacturing process where the root cause of the failure is coming from. The model shouldn't predict a false manufacturing process instead of predicting a true one. In other words, a wrong prediction that a failure is caused by a process (X) is totally rejected if it is actually caused by another different process. However, such a disadvantage can be improved during the transition stage where the process of automatic claims evaluation is running in parallel with the manual traditional one so as to improve the next trained model after a larger dataset size is accumulated.

**Table 4.** Confusion matrix for the model of severity ($M_S$).

| | **Class** | **1** | **2** | **3** | **4** | **5** | **6** | **7** |
|---|---|---|---|---|---|---|---|---|
| | | | | **Predicted Labels** | | | | |
| **True Labels** | 1 | 95% | 5% | - | - | - | - | - |
| | 2 | - | 89% | 7% | - | 4% | - | - |
| | 3 | - | 7% | 60% | 13% | 13% | - | 7% |
| | 4 | - | 17% | 3% | 67% | 3% | 7% | 3% |
| | 5 | - | 20% | 13% | - | 60% | 7% | - |
| | 6 | - | - | - | - | 12% | 88% | - |
| | 7 | - | - | 22% | - | 11% | - | 67% |

**Table 5.** Confusion matrix for the model of occurrence ($M_O$).

| | **Class** | **1** | **2** | **3** | **4** | **5** | **6** |
|---|---|---|---|---|---|---|---|
| | | | | **Predicted Labels** | | | |
| **True Labels** | 1 | 81% | 6% | 3% | 9% | - | - |
| | 2 | 14% | 48% | 14% | 10% | 10% | 5% |
| | 3 | - | 7% | 87% | 4% | 2% | - |
| | 4 | - | 9% | 9% | 78% | 4% | - |
| | 5 | - | - | 13% | - | 73% | 13% |
| | 6 | - | - | 5% | - | 5% | 90% |

**Table 6.** Confusion matrix for the model of impact ($M_I$).

| | **Class** | **1** | **2** | **3** |
|---|---|---|---|---|
| | | | **Predicted Labels** | |
| **True Labels** | 1 | 97% | 3% | - |
| | 2 | 18% | 80% | 2% |
| | 3 | - | 33% | 67% |

**Table 7.** Confusion matrix for the model of category prediction ($M_C$).

| | | Predicted Labels | | | | | | | |
|---|---|---|---|---|---|---|---|---|---|
| | **Class** | **A** | **B** | **C** | **D** | **E** | **F** | **G** | **H** |
| **True Labels** | A | 95% | - | - | - | - | - | 5% | - |
| | B | 14% | 86% | - | - | - | - | - | - |
| | C | - | - | 71% | - | - | 8% | 13% | 8% |
| | D | - | - | - | 83% | - | - | 17% | - |
| | E | - | - | - | - | 86% | - | 14% | - |
| | F | - | - | 25% | - | - | 63% | 13% | - |
| | G | 5% | 10% | - | - | - | 2% | 83% | - |
| | H | - | - | - | - | - | - | - | 100% |

Another approach to evaluate the developed models is to examine the RPN in the original dataset (actual RPN) against the resulted RPN from applying Equation (1) to the three predicted elements, call it (predicted RPN). The frequency histogram in Figure 8 compares the two readings (actual vs. predicted) for the overall dataset. The histogram shows a high overlapping of results between the two RPN values. Applying statistical accuracy measurements between actual and predicted values, resulted in a mean absolute error of 3.86 and a root mean squared error of 12.76 which both represent acceptable accuracy of predicted against actual. Therefore, this is another approach to evaluate the models developed and showing that these models are effective and efficient.

In contrast, this histogram in Figure 8 shows a shortage in predicting higher RPN values when the multiplication result is higher than 80 (the values larger than 140 in the histogram is a clear example). The reason behind this weakness is due to a lack of data at high classes for severity and occurrence in the training dataset. Enhanced accuracy can be reached by enlarging the training dataset and this could be fulfilled when more data is accumulated over time. Given that the dataset used in this activity contained 1532 claims for a single product in one year only. Further improvement can be achieved by reviewing the predictions of the models after a testing period, where an expert engineer can compare both the proposed AutoML approach with the traditional approach and conclude an enhanced and extended dataset that can be used for models retraining. Another approach to improve the models is to minimize the scale of classification for severity and occurrence from 1–10 to become 1–5 scale, such change will improve the model precision and accuracy. Hint, the accuracy for $M_I$ is higher than the others.

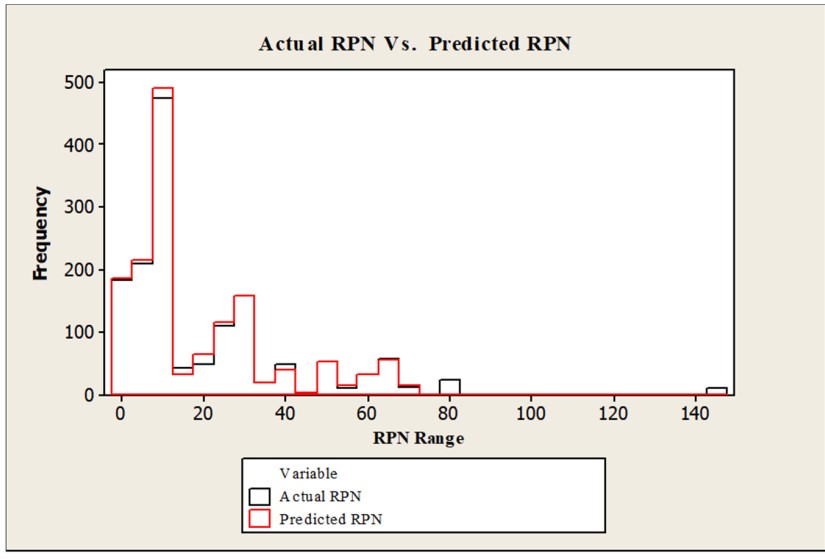

**Figure 8.** Originally evaluated RPN values frequency Vs Obtained RPN from the predicted Severity, Occurrence and Impact Classes.

Since the results of the proposed method are showing acceptable accuracy, given the dataset volume and used method, the models can be deployed at the partner company. The advantage of the proposed approach as compared to the traditional one is that it replaces the human intervention in the process and automates the decision-making process. In the traditional approach, once a claim is received from the mother company in Germany, a quality engineer in the quality management office in Hungary reviews the claim, decides the failure mode type and then assigns values to the three elements to calculate the RPN. Based on this judgment, further actions are decided. These actions can be by transferring the issue to critical issue resolution by using strategies such as the eight disciplines methodology (8D) if the RPN is above 160 points, or by updating the quality checklists at the production shop floor or could be both. However, this human intervention may imply some implementation error as it depends on the evaluator's experience. For example, assume a claim was evaluated by a quality engineer to be 160 points, while another engineer may underestimate the claim by ranking it to be 140 points based on his experience and memory. In the first situation, the engineer will transferee the claim to a more sophisticated process (8D strategy) which entails using more resources by forming a team to follow up and resolving the issue. On the other hand, the claim is highlighted to the production management (the second case). This is because such a process depends mainly on individual judgment and experience of the staff members who may give inaccurate estimation. Meanwhile, if such a process is done by a machine that makes decision-based on the accumulated leaning process, such uncertainty in decision making can be avoided. Thus, the proposed solution replaces this human intervention with a machine learning algorithm that evaluates claims based on the accumulated and none individualized experience and avoids the uncertainty in the experience of quality engineers. Moreover, the proposed approach can automatically analyze the new claims and construct correlations between incidents and therefore get a better ability for future prediction. Such process saves time, efforts, and improves responsiveness to failures either by alarming the quality management team instantly to serious issues or by automatically updating the quality checklists in the production shop floor by notifying labor and production staff of this issue in a real-time manner. From a business perspective, the proposed solution can be operated at any time and provides higher efficiency and effectiveness.

## 5. Conclusions

In this research, auto-machine learning was utilized to optimize failure modes handling by automatically identifying the failure mode, obtain its RPN and identify the manufacturing process related to the root cause of the issue. Three multiclass-classification machine learning models were developed to predict values for the RPN three elements namely severity, occurrence and impact. A fourth multiclass-classification model was developed to classify failures to their root cause process. The models' evaluation indicated relatively high accuracy models that can be deployed and integrated to enhance the company's ERP system. One of the features of the selected AutoML platform is its simple integration through the API, which is offered on the cloud. Such technology performs efficiently for large applications at the macro level of the factory. Utilizing such a solution enhanced the capabilities of the quality management team to handle any volume of claims data under high flow velocity. Such a solution will allow the quality team to focus on other strategic issues which will enhance the team's performance and results. The benefits of such technology do not end by this, but also could be furtherly extended to link claims and defects to the relevant manufacturing machine and operator. Once a claim is reported to the quality management it will be processed by the deployed model and instantly will be communicated to the relevant operators or managers and deeper to the shop floor in the factory. One more result for this research is that the manufacturing quality checklists for the selected product can be dynamically updated to include the top ten issues which are updated continuously according to their RPN. Such improvement enhanced the quality of processes and products. The factory in this study uses large screens on the shop floor to display quality checklists at every manufacturing process. These are used to review the quality issues while manufacturing processes are in place. A final check is being made at the quality gate of every process. The operators can watch the screens which

are updated every while and learn instantly about recently reported issues and making immediate correction actions. Finally, it is important to highlight the factors that impact the quality and accuracy of the developed models. For example, the accuracy of the model is strongly dependent on the quality of the data originated at the first point where the failure or defect was initially detected. Empty data rows, ambiguous information, or mistyping could forfeit important features and therefore, result in inaccurate prediction and reduce the system credibility. Therefore, a recommendation was suggested to the company to develop the data gathering platform (ERP system) in order to ensure higher quality prediction in the future. Furthermore, it is also essential to keep updating and maintaining the model by conducting periodical review sessions for the predicted RPN values and correct them when needed. Retraining the model using a larger volume of data will accumulate the model experience and improve model accuracy.

**Author Contributions:** S.S. (Conceptualization, Methodology, Investigation, Writing—Original Draft, Writing—Review & Editing). I.H. (Conceptualization, Supervision, Validation). M.D. (Supervision, Project administration). All authors have read and agreed to the published version of the manuscript.

**Funding:** This work was supported by the Stipendium Hungaricum Programme and by the Mechanical Engineering Doctoral School, Szent István University, Gödöllő, Hungary.

**Acknowledgments:** Special thanks to CLAAS Hungaria Kft, especially to Mr. Robert Csombordi, Head of Quality Management, for their endless support in conducting this research work.

**Conflicts of Interest:** The authors declare no conflict of interest.

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
