# Peer review of "Enhancing Failure Mode and Effects Analysis Using Auto Machine Learning: A Case Study of the Agricultural Machinery Industry"

_processes, doi:10.3390/pr8020224_

Round 1

Reviewer 1 Report

The authors developed machine learning based classification model to improve failure mode and effects analysis and the generation of risk priority number in auto industry. The authors use Google AutoML tool to develop the model but there is no explanation about the adopted algorithm. The performance of any machine learning model is governed by the type of the applied algorithms. The authors shall explain about the adopted algorithm and would be interesting if they compared its performance with other algorithms. In addition, the advantage of this approach compared with the traditional one shall be presented in discussion section in detail. There are few typos, e.g. line 366.

Author Response

Dear Sir,

Thank you very much for your valuable comments. We reviewed all of them and we modified our article accordingly.

We provided a point-by-point response in the attached file, please let us know if you have any further comments.

Thank you again for your efforts reading our work

Best regards

Authors

Reviewer 2 Report

The paper presents an approach to improving the FMEA method with the application of Auto-Machine Learning. The goal is to develop an automatic claim ranking system, which enhances the FMEA processes. The used dataset is a collection of failure incident reports over one year for the feeding house of a combine harvester. This set includes 23 different input features and 1532 data points. This data is used to train four different models to obtain four different results: Severity, Occurrence, Impact and the relevant manufacturing process. After the classification, the RPN is calculated and compared to a defined RPN from quality engineers.

In general, the writing style of the paper is very good. Most of the used words are easy to understand. The structure of the paper is very understandable and all pictures are big enough and easy to comprehend. However, the figures on page 4 could be in a higher resolution (Figure 1 and 2). The layout of the paper still needs to be partially revised, e.g. page 8 line 288 (table is hard to read) or page 10 line 348.

However, some issues are as follows:

The method itself has some issues. The first occurs in chapter 3.1. Data Pre-processing. The methods excludes severity scales over 7 and occurrence scales over 6. Consequently, the maximum RPN values are only up to about 150, but the whole range goes up to 300. Therefore, the whole process does not work for important and serious failures. However, exactly these failures need to be taken into account for the FMEA and therefore must be represented in the method. Without the use of these higher scales, a realistic application of the procedure in practice is only conceivable to a very limited extent. Another problem is the representation of the dataset. This should be explained briefly with the help of an example, so that on the one hand all input features are explained and on the other hand the content is easier to understand. Right now it only shows the top 5 input features and not the rest. The content of the dataset is not transparent either, as no information is provided in this paper. The chapter 3.2 Data Modeling is too short. It lacks important information about the used algorithms or methods. Furthermore, details like the used hyper-parameter of the different methods are not mentioned. Although no information about the used hardware or the calculation time is given. This chapter must be more specific about the used tools and method and not mention the ML Models like black boxes. Moreover, the paper title should be revised, because it says “A case study of Auto Industry” and the paper is about the agricultural industry (CLAAS). The results are well presented and clearly explained but the discussion part is too short. It misses to answer some important question like: What are the advantages of the method over a ruled based system? How can the accuracy be improved with the same dataset? How can the higher scales for severity and occurrence be taken into account, without generating more data? Why are these specific ML-Methods used for this problem?

Thus, a revision of the paper is suggested. Especially the revision of the method is important, so it can work with the higher scales with less data points. Furthermore, the ML-Methods should be explained in detail and more information about the dataset is necessary to understand the method.

Author Response

(The authors gave the same response as above.)

Reviewer 3 Report

The reviewed paper describes the application of data mining tools as a supporting for FMEA analysis of a production process. In the great majority real FMEA analyses expert opinions are required, and the collection of such opinions may be difficult and costly. The authors propose to use a machine learning approach where a multi-class classifier is built using data pre-processed by experts. This classifier may be used for automatic evaluation of future failures data. A publicly available machine learning tool, known as Automatic ML, and provided, e.g., by Google, is used for the construction of this decision support system. The authors describe a real application in a machine-building industry, where their system has been successfully implemented. The paper can be regarded as a case study describing the applicability of the proposed general approach.

The paper is interesting and well written but requires some changes with the aim to improve its applicability.

The description of input data is not sufficient. The majority of these data are of the textual character. It is not written if the processed texts are standardized allowing to process them in a categorical form, or if the interpretation of these texts is somewhat made by the used software. The proposed technique is based on a supervised classification methodology. In this methodology, input data are divided in two groups: attributes and evaluated classes. It is not explicitly written in the paper what are the classes, and who defined their values in the training and evaluation sets. One can guess that this is done by some experts (Fig.4 and the following text), and presented as Damage codes (Table 1). However, this information has to be presented in a more formal way known from machine learning literature. Experimental results have been presented using confusion matrices (Tab.4 – Tab.7). These matrices look somewhat differently from similar matrices presented in machine learning publications (see, Japkowicz, N. and Shah, M.: Evaluating Learning Algorithms. A Classification Perspective, Cambridge University Press,2015), and contain less important information. Suppose, for example, that frequent failures belonging to Category G trigger some costly actions. From Table 7 that this category is correctly indicated in 83% of cases when these actions are really needed. However, we do not know, what is the percentage of false alarms when Category G is incorrectly indicated by the used classifier. In my opinion, it is advisable to present both types of Confusion matrices in parallel (e.g, Table Xa and Table Xb). The title has to be changed. Auto machine learning is not a particular method of classification but type of a software tool. Therefore, classification cannot be based on AutoML but made using this type of IT tooll.

To sum up, the paper can be published after the introduction of rather minor changes proposed above.

Author Response

(The authors gave the same response as above.)

Round 2

Reviewer 1 Report

The authors address the provided comments.

Author Response

Dear Sir,

Thank you very much for your valuable comments and improvement recommendations.

your efforts are highly appreciated

Best regards

Authors

Reviewer 2 Report

Most of the mentioned suggestions of the revision have been implemented. The title has been revised, as has the chapter 3.2. Now it is easier to understand how the ML-model is constructed. Chapter 3.1 has also been adapted, especially Table 1. Now it is clearer to comprehend what data comes from where and what they contain. However, a fictional example would still be helpful to understand the whole dataset better. One stated question about how to improve the accuracy of the ML-process has not been addressed.

Furthermore, one important issue of the method was not revised: The applicability of the method for larger scales (higher 7 or 8) for the attributes “severity” and “occurrence”. Because without the use of these higher values, an application of the stated method is very limited.

At least an idea or concept should be presented in the conclusion section.

Author Response

Dear Sir,

Thank you very much for your valuable comments and improvement recommendations.

Kindly, please find the updated response documents for the second review round.

We highly appreciate your efforts and feedback

Best regards

Authors
